# OUT-DISTRIBUTION TRAINING CONFERS ROBUSTNESS TO DEEP NEURAL NETWORKS

**Mahdieh Abbasi, Christian Gagné**
Computer Vision and Systems Laboratory / REPARTI and Big Data Research Centre
Electrical and Computer Engineering Department, Université Laval, Québec (Québec), Canada
`mahdieh.abbasi.1@ulaval.ca, christian.gagne@gel.ulaval.ca`

## ABSTRACT

The easiness at which adversarial instances can be generated in deep neural networks raises some fundamental questions on their functioning and concerns on their use in critical systems. In this paper, we draw a connection between over-generalization and adversaries: a possible cause of adversaries lies in models designed to make decisions all over the input space, leading to inappropriate high-confidence decisions in parts of the input space not represented in the training set. We empirically show an augmented neural network, which is not trained on any types of adversaries, can increase the robustness by detecting black-box one-step adversaries, i.e. assimilated to out-distribution samples, and making generation of white-box one-step adversaries harder.

## 1 INTRODUCTION

Generalization properties of Convolution Neural Networks (CNNs) are remarkably good for some vision tasks such as object recognition. However, when a test sample is coming from a different concept that is not part of the training set, i.e. out-distribution samples, then CNNs force assignment to one of the classes of the original problem, possibly with high confidence. For example, while a given CNN trained on MNIST digits shows great accuracy on the corresponding test set ($> 99\%$), it maintains a confidence of $\approx 86\%$ on samples from NotMNIST dataset, which contains printed letters A-J. From a decision-making perspective, this is an issue, as the network shows a confidence that is clearly inappropriate.

Moreover, CNNs also suffer from adversarial examples artificially generated from clean samples, with the aim of fooling the model. To mitigate the risk of adversaries, detection procedures have been proposed for identifying and rejecting adversaries (Feinman et al., 2017; Metzen et al., 2017; Grosse et al., 2017; Abbasi & Gagné, 2017). For instance, Grosse et al. (2017) stated that adversaries are actually statistically different from clean samples. So one can augment the output of a CNN with an extra dustbin label (a.k.a. reject option), then train it on clean training samples and their corresponding adversaries (assigned to dustbin) in order to enable CNNs to detect and reject such adversaries. However, this assumes an access to a diverse set of training adversaries for identifying well the various types of adversaries. Feinman et al. (2017) used kernel density estimation in the feature space to identify adversaries, with mixed results given that some of adversaries become entangled with clean samples, which appears difficult to handle by kernel methods. However, instead of attempting to reject all adversaries, it seems better if a classifier can reject some adversaries as dustbin, while correctly classify others.

In this paper, we empirically show that an augmented CNN trained only on *natural out-distribution samples*, in addition to the problem training set, is able to reject some adversaries as dustbin while correctly classifying others. In fact, as such an augmented CNN reduces over-generalization in out-distribution regions, it learns a feature space that separates some of adversaries from in-distribution samples.

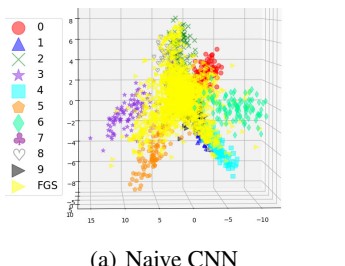
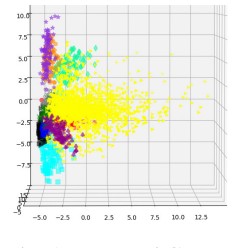

(a) Naive CNN      (b) Augmented CNN

Figure 1: Test samples from MNIST and their corresponding FGS adversaries plotted in a reduced feature space for (a) a naive CNN and (b) an augmented CNN. For visualization purposes, the dimensionality of feature space, which was achieved from the last convolution layer, is reduced to 3 using Principal Component Analysis (PCA). Refer to Fig. 3 in the appendix for more results.

## 2   OUT-DISTRIBUTION LEARNING FOR CNNs

It has been argued that a central element explaining the success of deep neural networks is their capacity to learn distributed representations (Bengio, 2009). Indeed, this allows such models to perform well in the regions that are only sparsely sampled in the training set, in possibly very high dimension space. However, neural networks make totally arbitrary decisions in the regions that are outside of the distribution of the learned concepts, leading to over-generalization. This can even provide some explanations on why naive neural networks incorrectly classify adversaries to task-related classes with high confidence.

Looking closely at the feature space of a CNN (i.e. output of the last convolution layer), we can see (Fig. 1(a)) that out-distribution samples are located close to the in-distribution samples in this trained feature space. To enable a CNN to be robust to out-distribution samples, we augment its output with an extra dustbin label, reserved for out-distribution samples. The augmented CNN is trained on both in-distribution and natural out-distribution samples. We observe (Fig. 1(b)) that such an augmented CNN is also able to disentangle the adversaries from in-distribution samples in the feature space, even though it is trained only on natural out-distribution samples.

## 3   EVALUATION

Using MNIST and CIFAR-10 datasets, we evaluate the augmented CNNs with black-box and white-box adversaries.

**MNIST vs NotMNIST**: Training on gray scale images of hand-written digits (MNIST dataset), using gray scale images of letters A-J (NotMNIST dataset[1]) as out-distribution samples.

**CIFAR-10 vs CIFAR-100**: Training on CIFAR-10, using samples from CIFAR-100's super-classes with no conceptual similarity as out-distribution samples – for more details, see the appendix.

### 3.1   BLACK-BOX ADVERSARIES

As black-box adversaries are transferable to other CNNs (Szegedy et al., 2013; Papernot et al., 2017), we generate T-FGS and FGS adversaries using cuda-convnet CNN (called GA-CNN), different from the models in our experiments (in terms of initial weights and architecture).

Comparison of naive CNNs with augmented CNNs in Table 1 shows that augmented CNNs maintain the same accuracy on clean MNIST test samples, while accuracy of augmented VGGs on clean CIFAR-10 test samples drop by $\approx 2\%$ in comparison to naive VGGs. Column "Dust" in Table 1 shows the percentage of adversaries rejected by the augmented CNNs. Although these networks are not trained on any types of adversaries, they can detect some of FGS and T-FGS adversaries as

---

[1]Available at `http://yaroslavvb.blogspot.ca/2011/09/notmnist-dataset.html`.

| Dataset | Model | Clean test Acc. | Adversaries by GA-CNN | | | | | |
| | | | FGS | | | T-FGS | | |
| | | | Acc. | Dust | Err. | Acc. | Dust | Err. |
| MNIST | Naive CNN | **99.50** | 35.14 | — | 65.86 | 19.99 | — | 80.01 |
| | Augmented CNN | 99.47 | 19.15 | 65.19 | **15.66** | 1.17 | 95.92 | **2.91** |
| CIFAR-10 | Naive VGG | **90.53** | 52.40 | — | 47.60 | 64.58 | — | 35.42 |
| | Augmented VGG | 88.58 | 45.02 | 30.46 | **24.52** | 49.12 | 39.06 | **11.82** |

Table 1: Performance on black-box adversaries attacks. two cuda-convnets were used for generating adversaries for MNIST and CIFAR-10. "Acc." corresponds to accuracy (the rate of correctly classified samples), "Dust" is the rejection rate, while "Err." is the misclassification rate (the samples that neither correctly classified nor reject as dustbin). All results reported are percentages (%).

| Dataset | Model | FGS | | T-FGS | |
| | | Success (%) | Distortion | Success (%) | Distortion |
| MNIST | Naive CNN | 91.91 | 0.187 | 94.48 | 0.170 |
| | Augmented CNN | **37.39** | 0.173 | **35.76** | 0.195 |
| CIFAR-10 | Naive VGG | 89.18 | 0.034 | 86.86 | 0.034 |
| | Augmented VGG | **49.28** | 0.035 | **58.93** | 0.036 |

Table 2: The success rates of generating white-box adversaries from naive and augmented networks. Two iterations are done for generating adversaries, with a step size of $\epsilon = 0.2$ and $\epsilon = 0.03$ for MNIST and CIFAR-10, respectively. "Success" corresponds to the rate of correct adversaries successfully generated within two iterations, while "Distortion" is the average distortion in the input space compared the original image ($L_2$ norm).

dustbin, while a proportion of them are correctly classified. This leads to a global error rate reduction of the augmented networks on adversaries through correct classification and rejection.

### 3.2 WHITE-BOX ADVERSARIES

As a second set of experiments, we generate white-box adversaries for MNIST and CIFAR-10 test samples from the same models tested in the previous section. To increase the success rate of generating adversaries, we applied FGS and T-FGS algorithms for two iterations (instead of one). Moreover, we forbid the choice of dustbin class as the fooling target class for T-FGS. However, such control for FGS algorithm is not possible since it is not targeted and as such we disregard the FGS adversaries with dustbin label — they are already recognized as such by the augmented networks.

As shown in Table 2, the augmented networks consistently have lower success rates of generating FGS and T-FGS adversaries. Increasing the number of iterations for the considered values of $\epsilon$ can raise the success rates, whereas this leads to adding more perturbations to clean images.

## 4 CONCLUSION

A key element allowing the rise of deep networks is their capacity to deal with high dimension spaces through distributed representation learning (Bengio & LeCun, 2007). However, deep networks incorrectly process instances from out-distribution regions by confidently classifying them as a predefined class, even though these instances are conceptually different from the trained concepts. Indeed, out-distribution samples are actually mapped very close to in-distribution samples in the feature space obtained from a naive CNN. In this paper, we argue that in-distribution concepts can be disentangled from out-distribution concepts through learning a more expressive feature space. To this end, we train augmented CNNs on natural out-distribution samples and in-distribution samples. Although the augmented CNNs are not trained on any kinds of adversaries, we empirically demonstrate that these CNNs separate some adversaries from in-distribution samples in their feature space. Therefore, these CNNs not only make some black-box adversaries detectable, but also make generating white-box adversaries harder.

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

# A APPENDIX

In this section, more details on the evaluation procedures as well as extra results are provided.

## A.1 DATASETS

### A.1.1 MNIST V.S. NOTMNIST

MNIST has training and test sets with 60K and 10K gray scale images contain hand-written digits, respectively. NotMNIST is a set of 18,724 gray scale images containing letters A-J. NotMNIST images have the same size as MNIST images, i.e. 28x28. We disregard the true labels of NotMNIST samples and instead these letter images are labeled as the dustbin class. Note, all the images are scaled to $[0, 1]$

For MNIST dataset, we consider a cuda-convnent architecture[2] that consists of three convolution layers with 32, 32, and 64 filters of 5x5, respectively, and one Fully Connected (FC) layer with softmax activation function. Each convolution layer is followed by relu, a pooling layer 3x3 with stride 2, and local contrast normalization (Hinton et al., 2012). In addition, dropout with $p = 0.5$ is used at the FC layer for regularization.

To train an augmented version of cuda-convnet, we create a training set comprising MNIST training samples along with 10K randomly selected samples from NotMNIST dataset. The remaining samples from NotMNIST ($\approx$8K) in conjugation with MNIST test samples are considered for evaluating the augmented CNN.

### A.1.2 CIFAR-10 V.S. CIFAR-100

Training and test sets of CIFAR-10 contain 50K and 10K RGB images with size 32x32. The classes of CIFAR-10 are airplane, automobile, bird, cat, deer, dog, frog, horse, ship, truck. As out-distribution samples for CIFAR-10, we consider CIFAR-100 dataset. To reduce a conceptual overlap between the labels from CIFAR-10 and CIFAR-100, we ignore super-classes of CIFAR-100 that are conceptually similar to CIFAR-10 classes. So, we exclude vehicle 1, vehicle 2, medium-sized mammals, small mammals, and large carnivores from CIFAR-100. Note, all the images are scaled to $[0, 1]$, then normalized by mean subtraction, where the mean is computed over the CIFAR-10 training set.

For CIFAR-10, we choose VGG-16 (Simonyan & Zisserman, 2014) architecture that has 13 convolution layers with filter size 3x3 and three FC layers. For regularization, dropout with $p = 0.5$ is used at FC layers. To train an augmented VGG-16, $30K$ randomly selected samples from the non-overlapped version of CIFAR-100 (labeled as dustbin class) are appended to CIFAR-10 training set.

## A.2 ADVERSARIAL ALGORITHMS

Broadly speaking, for a given input $\mathbf{x} \in \mathbb{R}^d$, an adversarial algorithm attempts to generate small and imperceptible distortion $\epsilon \in \mathbb{R}^d$ such that a victim classifier $h$ misclassifies the perturbed input $\mathbf{x}$, i.e. $h(\mathbf{x} + \epsilon) \neq h(\mathbf{x})$. In this paper, we consider one-step algorithms for generating adversaries.

**Fast Sign Gradient (FGS)** is proposed by Goodfellow et al. (2014) as a fast adversarial generation algorithm. Inspired from gradient descend algorithm, for each pair of a clean image and its associated true label, i.e. $(\mathbf{x}, y(\mathbf{x}))$, FGSM modifies the clean image $\mathbf{x}$ in order to maximize the loss function of the underlying classifier, i.e $\mathcal{L}$. Formally, a FGS adversary is generated as follows:

$$\mathbf{x}' = \mathbf{x} - \epsilon \times \text{sign}(\frac{\partial \mathcal{L}(h(\mathbf{x}), y(\mathbf{x}))}{\partial \mathbf{x}}), \tag{1}$$

where $\epsilon$ as the step size should be chosen large enough so that FGSM can generate adversaries after a single step.

---

[2]The configuration of this CNN is available at `https://github.com/dnouri/cuda-convnet/blob/master/example-layers/layers-18pct.cfg`

**Targeted Fast Sign Gradient (T-FGS)** generates adversaries to be misclassified into a selected target class ($y'$), which is different from the input's true label, i.e. ($y' \neq y(\mathbf{x})$) (Kurakin et al., 2016). For each sample, we select the least likely class according to the prediction provided by the underlying classifier. Using one iteration of gradient ascend algorithm, the loss function is maximized for a given pair of a clean image and a selected target class $y'$:

$$
\begin{aligned}
y' &= \arg\min\{p(h(\mathbf{x})|\mathbf{x})\} \\
\mathbf{x}' &= \mathbf{x} + \epsilon \times \mathrm{sign}(\frac{\partial \mathcal{L}(h(\mathbf{x}), y')}{\partial \mathbf{x}})
\end{aligned}
\tag{2}
$$

For generating *black-box adversaries* to attack other classifiers (such as augmented CNNs), we utilize cuda-convnet CNNs (called GA-CNNs) for both MNIST and CIFAR-10. Fig. 2 exhibits some clean samples with their corresponding FGS and T-FGS adversaries that are generated by GA-CNNs for MNIST and CIFAR-10.

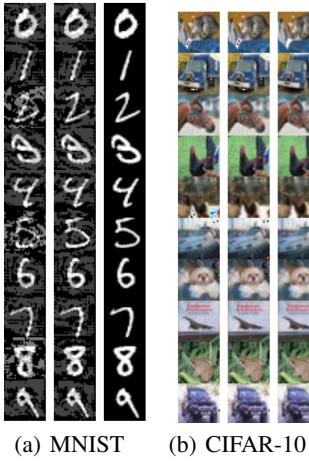

(a) MNIST     (b) CIFAR-10

Figure 2: First, second, and third columns present T-FGS, FGS, their corresponding clean samples, respectively.

### A.3 ADVERSARIES IN REPRESENTATION (FEATURE) SPACE

It is well known that the higher layers of deep neural networks extract more abstract features, which leads to transferring raw input data from the input space to an expressive feature space (Bengio et al., 2013; Bengio, 2009). Therefore, we used the last convolution layer of a CNN as a feature extractor in order to transfer the samples from the input space to feature space. In this section, we provide some visual results to compare the feature spaces of a naive CNN and an augmented CNN. For visualization purposes, the dimensionality of the feature spaces are reduced using PCA.

Fig. 3 illustrates MNIST samples and their corresponding adversaries in two feature spaces obtained from a cuda-convnet CNN and its output-augmented version. Note that while the cuda-convnet CNN trained on MNIST misclassifies NotMNIST samples with an average confidence $\approx 86\%$, the augmented cuda-convnet confidently classifies $99.96\%$ of them as dustbin.

Fig 4 presents feature spaces of VGG-16 and augmented VGG-16. While VGG-16 classifies out-distribution samples (i.e. CIFAR-100 samples) as one of CIFAR-10 classes with confidence $\approx 91\%$, augmented VGG-16 confidently classifies $95.36\%$ of these samples as dustbin, i.e. out-distribution samples.

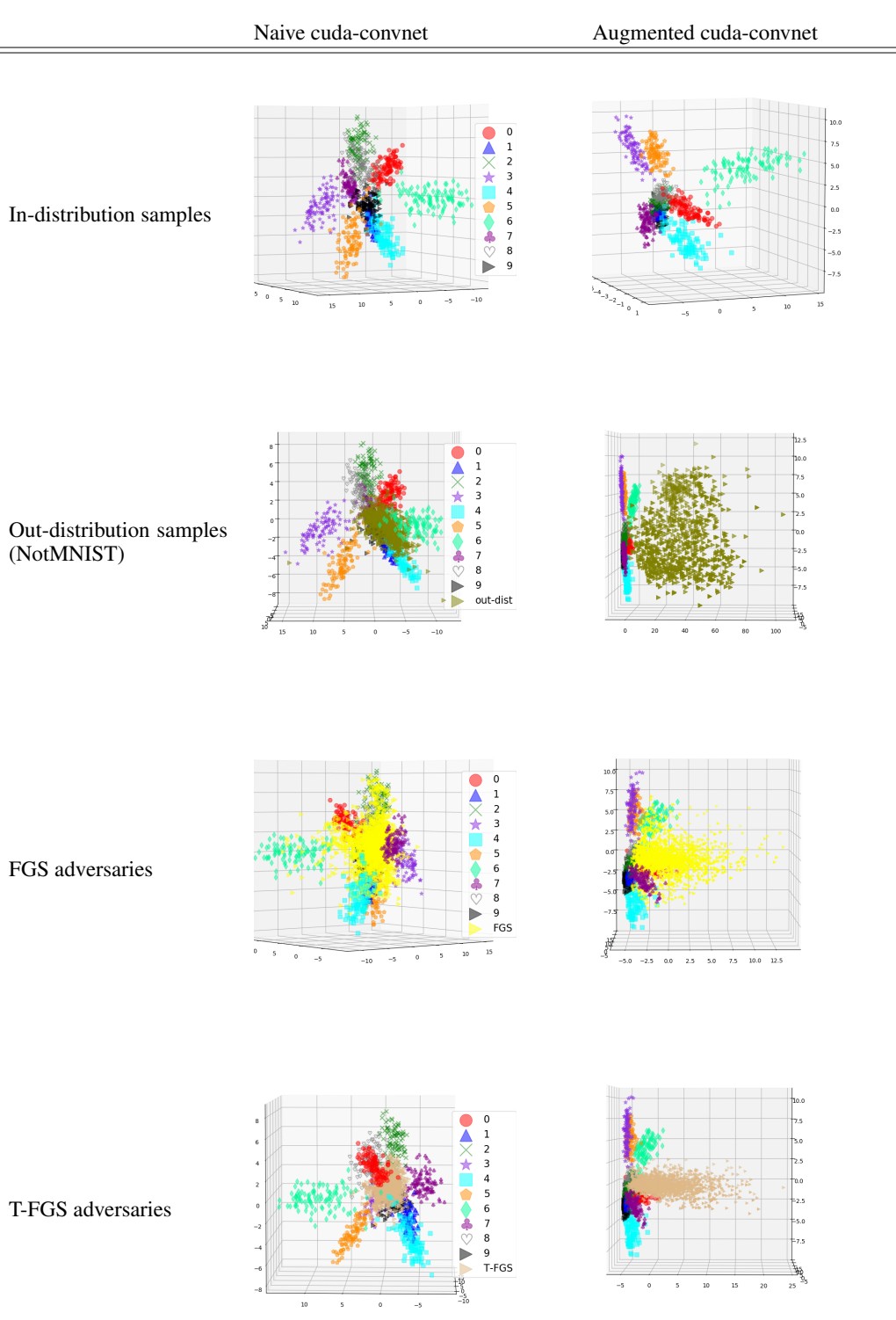

Table 3: Visualization of MNIST samples and their corresponding adversaries in the feature spaces learned by a naive cuda-convnet (first column) and an augmented cuda-convnet (second column).

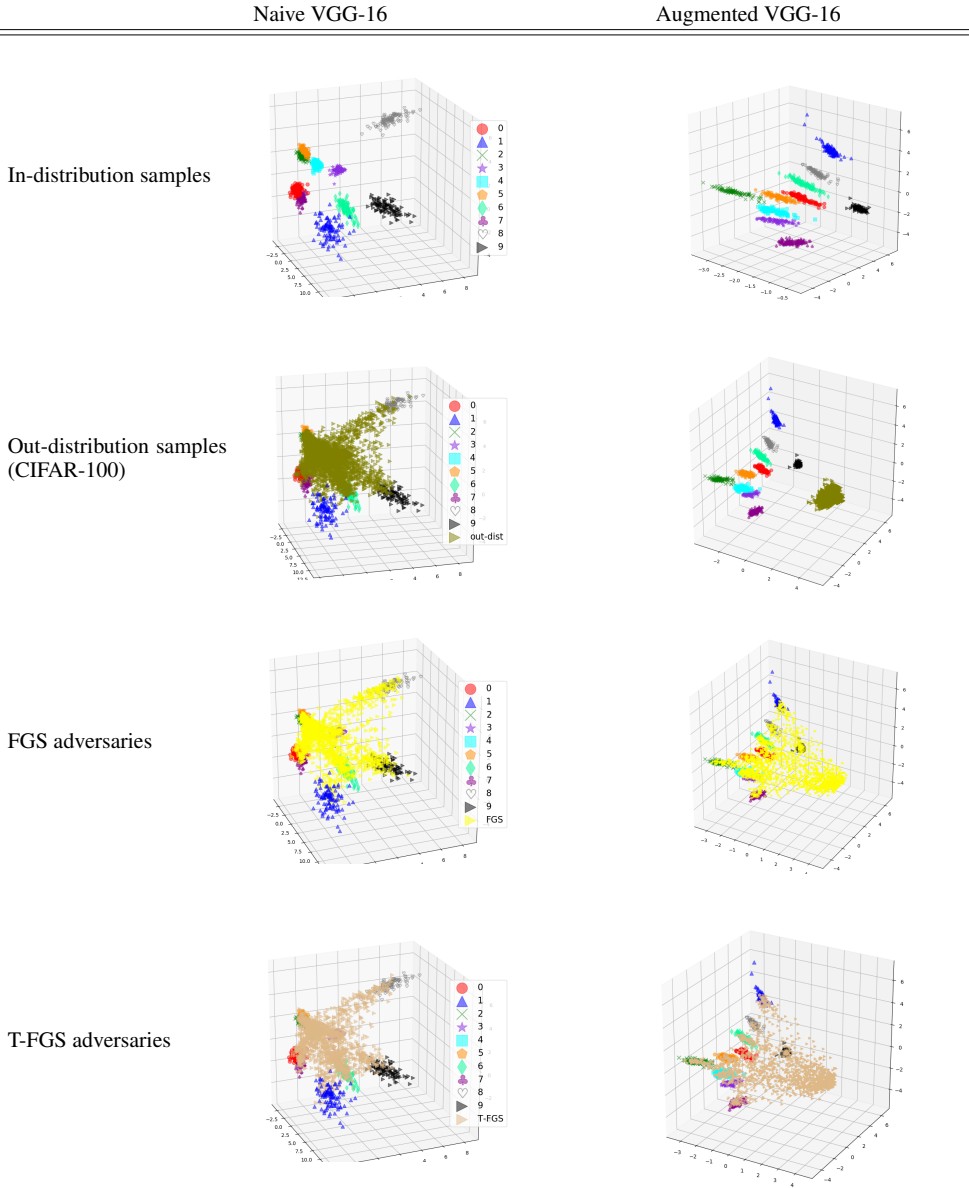

Table 4: Visualization of CIFAR-100 test samples and their corresponding adversaries in the feature spaces learned by a naive VGG-16 (first column) and an augmented VGG-16 (second column).

