# OpenReview forum: "Out-distribution Training Confers Robustness to Deep Neural Networks"
_ICLR.cc/2018/Workshop — Reject_

### Official Review · AnonReviewer2 · 2018-03-10
**interesting observation but missing experiments, related work**

**Rating:** 5
**Confidence:** 3

**Review:**

This paper makes a simple observation that my training a classifier with an additional "other" class, and utilizing data from outside the training set as examples for this class, the final trained classifier is more robust to fast gradient sign (FGS) adversarial examples than a similar classifier without the "other" class training.

Specific comments/questions:
- you begin referring to FGS and T-FGS without ever defining these terms or citing the work. Don't assume the reader knows the acronym for fast gradient sign, and regardless you need to cite this method. (I know this come up in the appendix, but this is too late, or at least direct the reader to appendix)
- similarly, cuda-convnet CNN is referred to without reference. I assume you mean AlexNet (cuda-convnet  is an old and non-standard way of referring to this architecture), but you need to say this. Same goes for citation of VGG net.
- table 1 is very confusing. What is the accuracy column referring to. It is clearly not accuracy of the test images since that is given by column"clean test acc". The other possibilities I can see is are (i) the accuracy of the dustbin classifier (i.e. how frequently it correctly classifies an adversarial example as "other" or (ii) the accuracy of overall network on adversarial examples where a classification is considered correct if either an image is classified as "other" or classified correctly as one of k real classes. In both these cases a higher accuracy is better, but the augmented network does worse on both cifar and mnist.  Please specify what this column refers to and also make clear in paper.
- How does your method work against other attack methods? There are so many easy to run toolboxes for generating adversarial examples, I don't think there is any excuse for not running your approach with other attacks.
- How does this idea relate to other related work.. for example, GAN models have been proposed where the discriminator classifies 1 of k classes (trained with real data) and also classifies generated images as "other".. that seems akin to this approach, but with the out of distribution samples coming from a generator rather than a fixed real dataset. Similarly, what happens if you run adversarial training methods with the same model architectures (i.e. generate adversarial examples and classify them as "other"). How does this compare to other methods of classifying out of distribution examples, and can any of them be used to classify adversarial examples?
- what happens when the out of distribution dataset is varied? it would be interesting to see a couple different datasets used here.
- what happens if you utilize the out of distribution samples in another way, e.g. asking the network to have uniform output distribution over predicted classes?

The main observation in the paper is interesting, but the experiments are lacking, relation to other work is missing and overall I don't come away from the paper understanding *why* their results show what they do. I would increase my score if the authors can address the questions above.

---

### Official Review · AnonReviewer1 · 2018-03-10
**unconvincing results. robustness is a too strong a claim!**

**Rating:** 5
**Confidence:** 4

**Review:**

I think the paper is well written, clear and on one of the biggest problems in the field. I cannot perfectly assess originality but I find the approach limited in scope and potential impact.

Knowing that a neural network can shatter a dataset in arbitrary ways (e.g. learn a classifier to random labels) I find it highly unlikely that adding a small set of points from another fixed distribution whose distance to the current one is uncontrolled would help alleviate the problem much less "confer robustness". I don't see anything in the paper to convince me otherwise.

pros:
- simple method that seems to help somewhat on the datasets they have tested on

cons:
- even for cifar10 the reduction seems too small and will probably diminish further if scaling it up to ImageNet
- the idea seems adhoc and unconvincing

---

### Decision · Program_Chairs · 2018-03-20
**ICLR 2018 Workshop Acceptance Decision**

**Decision:**

Reject

**Comment:**

Based on the reviews, this paper has not been accepted for presentation at the ICLR workshop. However, the conversation and updates can continue to appear here on OpenReview.